# Tight Integration of GNSS and Static Level for High Accuracy Dilapidated House Deformation Monitoring

**Jian Yang** [1], **Weiming Tang** [2] , **Wei Xuan** [1,]*  and **Ruijie Xi** [1]

1 School of Civil Engineering and Architecture, Wuhan University of Technology, 122 Luoshi Road, Wuhan 430070, China; tjxyyj@whut.edu.cn (J.Y.); rjxi@whut.edu.cn (R.X.)
2 GNSS Research Center, Wuhan University, 129 Luoyu Road, Wuhan 430079, China; wmtang@whu.edu.cn
* Correspondence: xuanwei1988@whut.edu.cn; Tel.: +86-133-4983-7800

**Abstract:** Global Navigation Satellite System (GNSS) can provide high-precision three-dimensional real-time or quasi-real-time changes of monitoring points automatically in house monitoring applications. However, due to the signal sheltering problem, large observation noise and multipath effects in urban observing environment with dense buildings, ambiguity resolution would be hard, and GNSS accuracy cannot always achieve millimeter level to satisfy the requirement of house monitoring. Static level is a precision instrument for measuring elevation difference and its variations, with a precision up to sub-millimeter level. It could be integrated with GNSS to improve the positioning accuracy in height direction. However, the existing integration of GNSS and static level is mostly on a respective results level. In this study, we proposed a method of integrating GNSS and static level observations tightly to enhance the GNSS positioning performance. The hardware design and integration mathematic model in data processing were introduced, and a group of experiments were carried out to verify the performance in positioning with and without the static level observation constraints. It found that the vertical monitoring measurement results of static level can achieve less than 1 mm. The GNSS ambiguity resolution performance can be improved by incorporating the measurement of static level into GNSS positioning equation as external constraints, and the precision of GNSS float solutions was significantly improved. Finally, the static level constraint can further improve the accuracy of the fixed solution from about 2 cm to better than 2 mm in vertical direction, which is even better than the accuracy in horizontal directions with about 3–6 mm with the static level constraint. The tight combination data processing algorithm can significantly improve the working efficiency, accuracy, and reliability of the application of dangerous house monitoring.

**Keywords:** deformation monitoring; GNSS; static level; tight combination

## 1. Introduction

Housing safety is directly related to people's basic life. With the expansion of the urban fringe and the explosion of the population, the pace of the elimination of traditional old housing has gradually accelerated [1]. At present, the old and dilapidated houses in cities and towns are mostly built earlier, and the main structure of the buildings were mostly composed of masonry and concrete. Due to the influence of external natural conditions, such as geological activities, weather conditions, and environmental variations, the structural strength of the house clearly attenuates under long-term use [2]. Wall cracks and tilts can even be observed in the case of some dilapidated and dangerous houses. Therefore, in the process of maintenance and demolition of these old buildings, proper treatment and safety monitoring should be carried out throughout the whole process. Traditionally, the supervision and management of the old and dilapidated houses were carried out by artificial inspection. Because the dilapidated houses generally scattered distributed, the traditional manual inspection method is inefficient and has limitations in the management level and comprehensiveness. It is hardly to provide 7/24 continuous

monitoring, and the main structure of the building and local key stress points can only be inspected periodically [3]. Some monitoring methods, such as ray and echo method, cannot reflect the real condition of dilapidated house comprehensively because of the limited monitoring range and of the randomness. It is easy to cause large errors in statistical analysis and hazard classification. Meanwhile, the traditional ways need to consume manpower and material resources and other supervision costs, and the supervisory staff must have professional knowledge and rich experience.

To realize the automatic management of dangerous house monitoring, Yu et al. proposed an architecture of dangerous house health monitoring system based on cloud platform [3]. Gao et al. established a dynamic monitoring system and management platform for dangerous houses by using a dip angle sensor and a settlement sensor combined with manual inspection [1]. By integrating a magneto strictive displacement sensor, a dip Angle sensor and the wireless communication technology, a kind of special sensor equipment for dangerous house monitoring is developed, which can realize unattended permanent measurement and achieve high precision [4]. These studies have made some achievements in the dangerous house monitoring. However, efforts still need to make in the monitoring comprehensiveness and accuracy, especially in the measuring of horizontal position and height difference between two points which are far from the main structure of the dilapidated house [5].

As a spatial geodesy technology, Global Navigation Satellite System (GNSS) has the advantages of all-weather, automatic, and real-time working ability, and the accuracy can achieve to millimeter level in horizontal and vertical directions [6]. The GNSS positioning technology has been widely applied in the displacement monitoring of dams, bridges and tall buildings etc. Hristopulos et al. applied accelerometer and GPS technologies to extract the dynamic response characteristics of a tall building under the wind load [7]. M. Evers et al. designed a GPS-based dam deformation monitoring system to monitor the displacement of Paraeiros Peiros dam in the service period [8]. However, due to the satellite geometry in the vertical component is not as strong enough as that in the horizontal component, the positioning accuracy in the vertical component can only achieve up to 3–5 mm level. Small displacements at 1–2 mm level cannot be detected by the GNSS technology only. To improve the positioning accuracy and stability of GNSS deformation monitoring, Yang et al. proposed a standard Kalman filter and robust estimation combination method to effectively resist the monitoring data gross error [9]. A number of studies proposed the combination of GNSS and static level observations to improve the vertical monitoring precision, and the method has been applied to monitor the foundation settlement and the deformation of the offshore oil platform [10–13]. However, at present, the two sensors were separated in the data processing model. The satellite navigation system only provided deformation data in horizontal direction, while the static level provided the vertical displacements. Although this monitoring mode have realized the high precision 3D deformation monitoring of buildings to a certain extent, it did not fully utilize the advantages of the observation integration of the two technologies. For instance, under the harsh observing environment in urban, GNSS satellite signals are easily obstructed by trees and buildings [14,15]. The positioning precision of GNSS would be decreased and unstable, especially in the vertical direction. If the static level and the GNSS are integrated at the observation level, the positioning accuracy and reliability of GNSS could be effectively improved, thus playing a greater role in the application of dangerous house monitoring [16].

In this paper, we proposed a static level and GNSS integration method in the observation level. The combination method in data processing model and hardware will be introduced in Sections 2 and 3. An experiment will be conducted in Section 4, and the performances of this method are shown in Section 5. At last, we give the conclusion in Section 6.

## 2. Instrumentations

### 2.1. Static Level

Static level is also called connected pipe level. The liquid level monitoring value in the static level container is constantly obtained during use. When the settlement of the monitoring point occurs, the liquid height in each container changes, and the liquid level difference in each container is measured by the displacement sensor so as to directly measure the settlement value of the monitoring point. According to the principle that the liquid level of the storage tank connected with the pipe is always kept level, the relative difference monitoring settlement of each static level is calculated by measuring the liquid level height of different liquid storage tanks, as shown in Figure 1. When the liquid density in the container and the external environment is the same, the liquid level in the container is at the same height, h1, h2, . . . , hn, as shown in Figure 1a. When vertical displacement occurs at the monitoring point, the liquid level inside the container is adjusted to form a new height of the same liquid level. The readings are h1′, h2′, . . . , hn′. Thus, the variations are △h1 = h1′ − h1, △h2 = h2′ − h2, . . . , △hn = hn′ − hn. Based on this method, the height movements of the monitoring station can be obtained by △H2 = △h1 − △h2, . . . , △Hn = △h1 − △hn. The liquid static level has the advantages of high accuracy and stability, and the accuracy can achieve up to 0.01 mm.

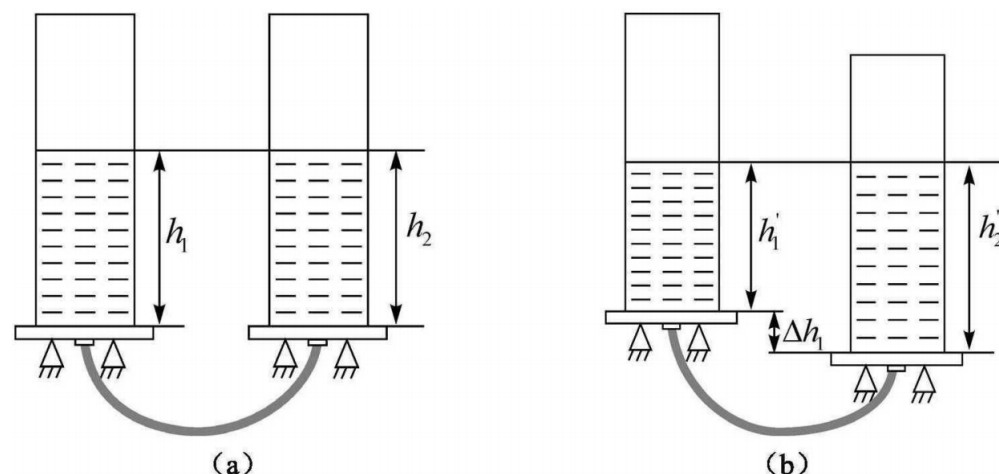

**Figure 1.** The principle of the static level. The original stage (**a**) and the new statement when the vertical displacement occurs (**b**).

Static leveling system based on static level is mainly used for monitoring vertical displacement and tilt of dams, nuclear power plants, high-rise buildings, mines, landslides, bridges and so on. The static leveling system is generally installed on the measuring pier or wall contour line of the monitoring object. The integrated modular automatic measuring unit is usually used to collect data, which is connected with the computer through wired or wireless communication, so as to realize automatic observation.

Compared with the total station and the geometric leveling method, the static leveling method has the characteristics of high precision, suitable for long-term monitoring mode and multi-purpose automatic monitoring. It is flexible in site setting and suitable in the long-distance deformation monitoring application of tunnels, railways and dams etc. At the same time, static leveling can be measured in real-time, which has high application value in the settlement monitoring during engineering construction.

### 2.2. Composite of Static Level and GNSS Receiver

In order to realize the tight combination of GNSS and static level, it is necessary to integrate the hardwires to ensure the alignment of GNSS antenna center and energy level center in space, so that the GNSS antenna and static level center are in the same plumb line. In this study, we designed an industrial chassis to integrate the GNSS and static level,

and the equipment has waterproof, shockproof and other industrial performance. The mechanical design drawings of this equipment are shown in Figure 2, and the detailed appearance diagram is shown in Figure 3.

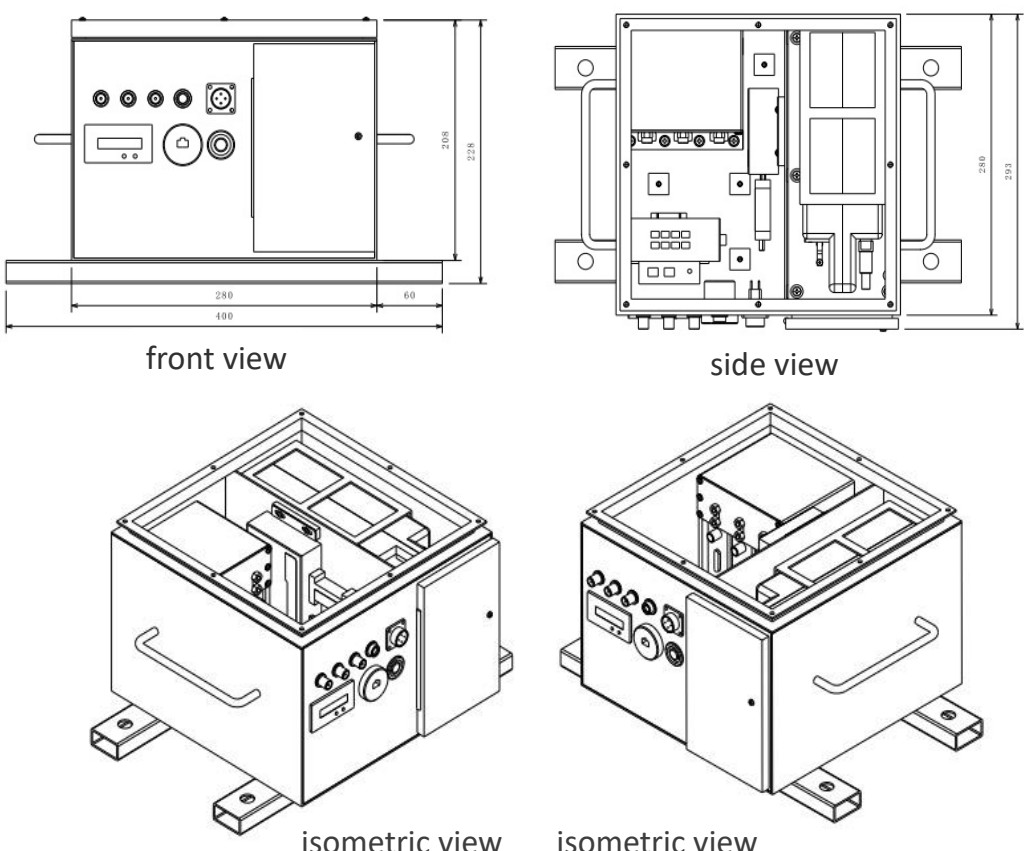

**Figure 2.** The mechanical design drawings of the GNSS antenna and static level integration equipment.

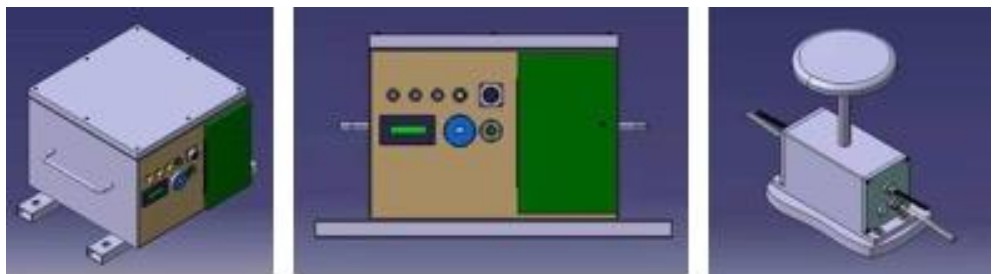

**Figure 3.** The appearance diagram of the GNSS antenna and static level integration equipment.

The static level data output signal is based on MODBUS-RTU protocol, and the data is outputted by a RS485 serial port. The time synchronization is completed with the time label of the GNSS module by sending the request periodically and collecting data for 1 s. In the static levelling system, multiple static levels are arranged into series to collect the settlement of multiple monitoring stations. The data collected by each static level is collected to the central processor through RS485 bus. The outputs of GNSS receivers are based on RS232 serial port which is collected to the central processor. In this study, three static levels were integrated with three antennas which connected with one receiver, and each set of static level corresponded to one GNSS antenna. The system structure drawing is shown in Figure 4.

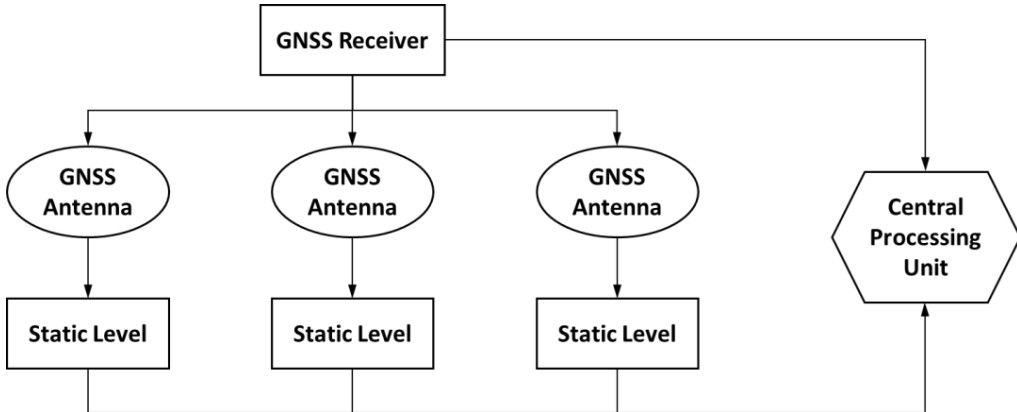

**Figure 4.** System structure drawing.

The GNSS receiver used in this paper is Panda Receiver PD318, and the static level devices are also designed and produced by Panda Company. In this system, GNSS positionings were used to provide the horizontal displacement monitoring and the absolute vertical coordinate of monitoring points, while static level provided the relative height changes of monitoring points. For the house safety monitoring, the tilt of the structure can be calculated through the horizontal displacement and the height changes of monitoring points. In practical applications, the outputs of this system can be further connected to GNSS-CORS (Continuous Operating Reference Station system) signals to obtain the absolute location information of monitoring points.

## 3. Methods

The tight combination of GNSS and static level can make up the disadvantage of GNSS elevation accuracy. At the same time, the static leveling results can provide vertical constraint for GNSS double difference ambiguity solution and assist the fast search and fixing of ambiguity.

In a deformation monitoring case, the baseline to be measured is *PQ*, where *P* is the reference point, and the point *Q* is the undetermined monitoring point. Then, the observation equation of GNSS single epoch pseudorange and carrier phase observations can be written as follows:

$$L_R = B \cdot \begin{bmatrix} dX \\ dY \\ dZ \end{bmatrix} + \varepsilon_R \tag{1}$$

$$L_\varphi = \begin{bmatrix} B - \lambda_G I_{n-1} \end{bmatrix} \cdot \begin{bmatrix} dX \\ dY \\ dZ \\ N_{(n-1)\times 1} \end{bmatrix} + \varepsilon_\varphi \tag{2}$$

where *B* is the coefficient matrix of the double difference equation; $\lambda_G$ is the wavelength of GNSS signals; $I_{n-1}$ is the unit matrix; $\begin{bmatrix} dX & dY & dZ \end{bmatrix}^T$ is the three-dimensional baseline vector; $N_{(n-1)\times 1}$ is the ambiguity vector. Considering that short baselines are often used in the deformation monitoring applications, troposphere and ionosphere errors can be considered to have been fully eliminated in the double difference model. Therefore, they are ignored here.

As for the height difference measurement of the static level, its measurement accuracy is high enough. Thus, we treat it as a constraint condition to join it in the GNSS double difference observation equation. In order to convert height difference components into the GNSS 3D baseline vector parameters, a vertical station centered cartesian coordinate system should be established. As shown in Figure 5, a left-hand coordinate system is built with station *P* as the origin, the vertical line of point *P* as the axis *H* (positive to the zenith),

the meridian direction as the axis $X$ (positive to the north), and the $y$ axis is perpendicular to the $x$ and $H$ axes (positive to the east).

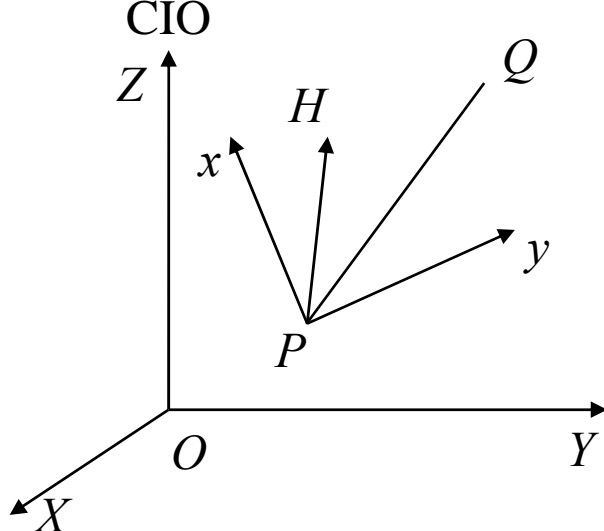

**Figure 5.** The left-hand coordinate system.

If the longitude and latitude of station $P$ is $(\lambda, \varphi)$ in the geodetic coordinate system, the baseline of $PQ$ in the local cartesian coordinate system would be

$$\begin{bmatrix} x \\ y \\ H \end{bmatrix}_{PQ} = \begin{bmatrix} -\sin\phi\cos\lambda & -\sin\phi\sin\lambda & \cos\phi \\ -\sin\lambda & \cos\lambda & 0 \\ \cos\phi\cos\lambda & \cos\phi\sin\lambda & \sin\phi \end{bmatrix} \begin{bmatrix} X_Q - X_P \\ Y_Q - Y_P \\ Z_Q - Z_P \end{bmatrix} \tag{3}$$

where $\begin{bmatrix} x & y & H \end{bmatrix}^T_{PQ}$ is the coordinate difference of $PQ$ in the local cartesian coordinate system; $X_Q, Y_Q, Z_Q$ and $X_P, Y_P, Z_P$ are the coordinate of $Q$ and $P$ in the geocentric coordinate system, respectively.

The vertical component could be extracted from (3)

$$dH = \begin{bmatrix} \cos\varphi\cos\lambda & \cos\varphi\sin\lambda & \sin\varphi \end{bmatrix} \cdot \begin{bmatrix} dX \\ dY \\ dZ \end{bmatrix} + \varepsilon_H \tag{4}$$

where $dH = H_{PQ}$, $\begin{bmatrix} dX \\ dY \\ dZ \end{bmatrix} = \begin{bmatrix} X_Q - X_P \\ Y_Q - Y_P \\ Z_Q - Z_P \end{bmatrix}$.

Considering that in a small measurement area, the variation difference between the earth height difference and altitude difference at two points can be ignored. Therefore, considering (1), (2) and (4), we can have

$$\begin{bmatrix} L_R \\ L_\varphi \\ dH \end{bmatrix} = \begin{bmatrix} B & 0_{(n-1)\times(n-1)} \\ B & -\lambda_G I_{n-1} \\ B_1 & 0_{3\times(n-1)} \end{bmatrix} \cdot \begin{bmatrix} dX \\ dY \\ dZ \\ N_{(n-1)\times 1} \end{bmatrix} + \begin{bmatrix} \varepsilon_R \\ \varepsilon_\varphi \\ \varepsilon_H \end{bmatrix} \tag{5}$$

where $B_1 = \begin{bmatrix} \cos\varphi\cos\lambda & \cos\varphi\sin\lambda & \sin\varphi \end{bmatrix}$.

If the weights of the pseudorange, carrier phase observations and the external damping coordinate of $H$ are $P_o = diag(P_R, P_\varphi, P_H)$, the corresponding coefficient matrix of the normal equation is

$$N_a = \begin{bmatrix} B^T P_R B + B^T P_\varphi B + P_H & B^T P_\varphi \lambda \\ P_\varphi B\lambda & P_\varphi \lambda^2 \end{bmatrix} \tag{6}$$

while the coefficient matrix of the normal equation without the constraint is

$$N_a = \begin{bmatrix} B^T P_R B + B^T P_\varphi B & B^T P_\varphi \lambda \\ P_\varphi B \lambda & P_\varphi \lambda^2 \end{bmatrix} \tag{7}$$

By comparing Equations (6) and (7), we know that there is one more item in the upper-left submatrix of Equation (6) than that of Equation (7). In the case of a single epoch, the observation ability of the design matrix is poor, and the upper-left submatrix of Equation (7) is prone to ill-condition, which is also the reason why it is difficult for the conventional On The Fly (OTF) method to solve the ambiguity in a single epoch. Adding a diagonal matrix independent to the coefficient matrix will greatly improve the ill-condition property of Equation (7) and improve the accuracy of the floating-point ambiguity resolution. Thus, the least-squares ambiguity decorrelation adjustment (LAMBDA) method could be applied to search and fix the ambiguities.

The method mentioned above could be called as the damped LAMBDA algorithm with external constraints [6,17,18]. As for the kinematic deformation monitoring applications, the damped LAMBDA algorithm could improve the precision of float ambiguity estimations, which is conductive to the fast ambiguity resolution for the single frequency GNSS positioning. The positioning reliability would be further improved by combining multi-frequency and multi-GNSS observations. Meanwhile, the external constraints could improve the overall positioning solution precision in terms of the multi-epoch static deformation monitoring mode. When the ambiguities were correctly fixed, the observation equation of (5) can be expressed as:

$$\begin{bmatrix} L_R \\ L_\varphi + \lambda_G \cdot N \\ dH \end{bmatrix} = \begin{bmatrix} B \\ B \\ B_1 \end{bmatrix} \cdot \begin{bmatrix} dX \\ dY \\ dZ \end{bmatrix} + \begin{bmatrix} \varepsilon_R \\ \varepsilon_\varphi \\ \varepsilon_H \end{bmatrix} \tag{8}$$

In this way, the 3D baseline resolution can be estimated with (8).

## 4. Results

### 4.1. Experiment Design and Data Collecting

We carried out an experiment on the roof of a residential building over 100 m in Wuhan, China. The observation environment on the roof was complex, as shown in Figure 6. GNSS signals could be sheltered by surrounding buildings, which was common with the general dangerous building monitoring environment. Three monitoring sites (called site A, B and C) were set to do the experiment, and a height adjustment device with sub-millimeter accuracy level was equipped under the GNSS antenna of site B.

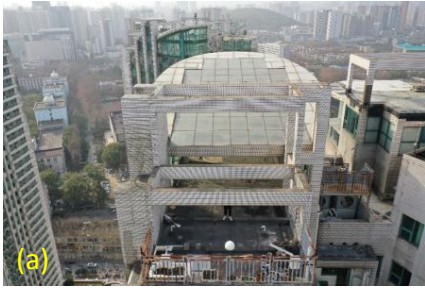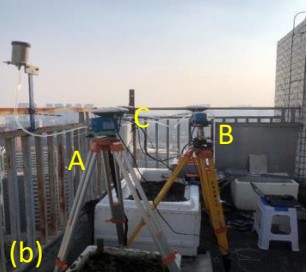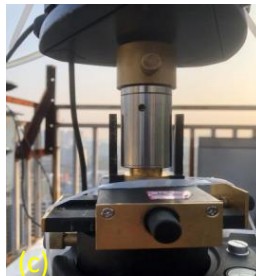

**Figure 6.** (**a**) Observing environment; (**b**) site settings; (**c**) height adjustment device.

The experiment was carried out as follows:

**Session 1**: At the original statement, four groups of data were collected, and the duration time was five minutes for each group.

**Session 2**: The height adjustment device at B was set up 2 mm. Then, four groups of data were collected, and the duration time was five minutes for each group.

**Session 3**: The height adjustment device at B was set up 3 mm. Then, four groups of data were collected, and the duration time was five minutes for each group.

**Session 4**: All the facilities were set power off and power on again to collect data for four groups. The duration time was five minutes for each group.

**Session 5**: The height adjustment device at B was set down 3 mm. Then, four groups of data were collected, and the duration time was five minutes for each group.

**Session 6**: The height adjustment device at B was set down 2 mm. Then, four groups of data were collected, and the duration time was five minutes for each group.

As previous mentioned, the sampling rate of static level is 1 Hz. The GNSS data sampling rate is 1 Hz as well. The GNSS data were processed with a homemade software called GNSStrack and the detailed GNSS data processing strategies were shown in Table 1.

**Table 1.** GNSS data processing strategies.

| Items | Strategies |
| --- | --- |
| Observations | GPS C1/P2/L1/L2 and BDS P1/P2/B1/B2 |
| Parameter estimation | Least squares |
| Cut-off elevation | 10° |
| Sampling rate | 1 Hz |
| Session length | 5 min |
| Weight method | Elevation dependent weighting method |
| Ambiguity Resolution | Searched and fixed by LAMBDA for every epoch |
| Tropospheric delays | Eliminated by double-differenced method |
| Ionospheric delays | Eliminated by double-differenced method |

### 4.2. Static Level Data Analysis

Figure 7 shows the height distance variations of B and C toward A provided by the static level. As can be seen from Figure 6, the measurement accuracy of static level in the vertical direction can achieve to sub-millimeter magnitude, and the measurement results are stable and reliable. More importantly, it is basically not affected by the external environment, so it can be used as a high-precision constraint condition to assist GNSS solution.

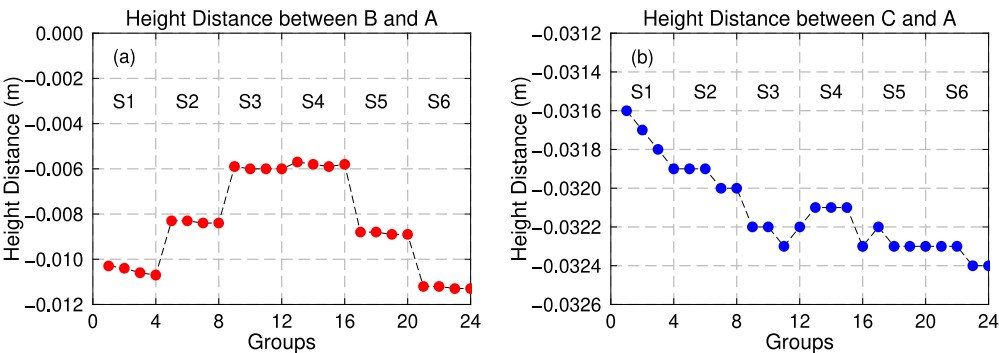

**Figure 7.** Height distance variations of B (**a**) and C (**b**) toward A provided by the static level.

### 4.3. GNSS Data Quality Analysis

In urban observation environment with dense buildings, the quality of GNSS data has a crucial impact on the monitoring accuracy of dangerous houses. The analysis and evaluation of GNSS data quality is beneficial to the data processing and algorithm development and design, and to accurately evaluate the effectiveness of various processing methods [19].

Translation, Editing and Quality Checking (TEQC) is an open free software for the data management service of GNSS monitoring station in geological research [20,21]. It is jointly developed by American satellite navigation system, crustal deformation monitoring and universities. It is recognized as one of the GNSS measurement data preprocessing software which is powerful, simple and easy to use. The software can calculate the multipath effect, ionospheric delay change rate [22] and other influencing factors through the combination

of various GNSS observation values to check the quality of data. The quality evaluation indicators of GNSS observation data can be generally divided into three categories: indicators reflecting the influence of errors, indicators reflecting the completeness of observation data and indicators related to the position of satellites [23–25]. The indicators that reflect the influence of the error mainly include: receiver clock difference, Signal-to-Noise Ratio (SNR), multipath effect, number of skip observations (O/SLPS), etc. The indexes related to satellite position mainly include satellite azimuth, altitude angle and geometric distribution. In order to display GNSS observation quality graphically, we developed an interactive interface for visual quality inspection software based on TEQC kernel in Figure 8. Table 2 shows the indicator statistics of the SNR, the completeness of observation data, multipath effect and the O/SLPS, and Figure 9 shows the time series of SNRs of one data case. It shows that, the SNR time series of GPS PRN 8 and PRN 14 were at a relative lower level compared with other satellite. The observation noise may larger than other satellites.

According to the data analysis results of the inspection software in Table 2 and Figure 9, all the observed data have warnings in varying degrees, indicating that in the urban environment, GNSS signals are affected by occlusion, interference and multipath effect. Therefore, it is difficult to obtain the ideal "clean" observation data. This could have a negative effect on the high precision positioning solution. Figure 10 shows the skyplot of site A. Due to the signal sheltering by buildings, only five GPS satellites can be observed during the experiments and most of BDS satellites were located on the south side of the station. In this case, the positioning precision in north-south and up directions would be lower than that of the east-west direction.

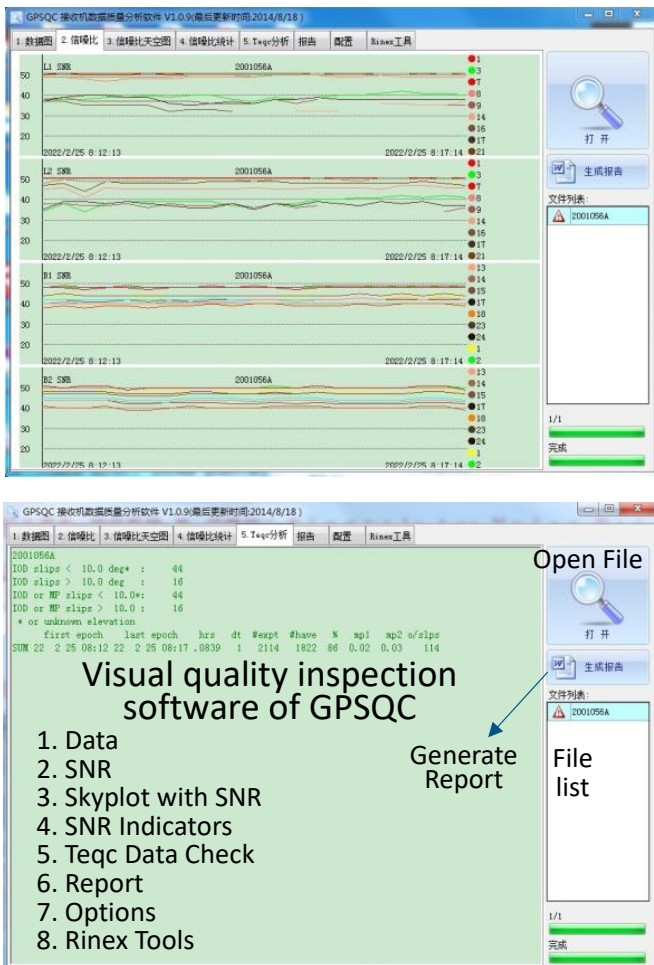

**Figure 8.** Interactive interface of the visual quality inspection software.

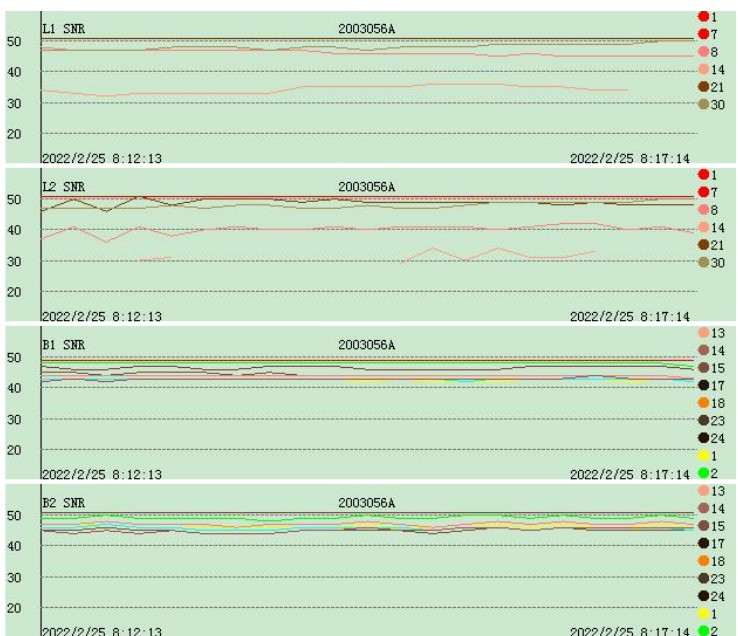

**Figure 9.** SNR time series of a data case.

**Table 2.** Indicator statistics of SNR, completeness of observation data (COD), multipath effect (MP) and the O/SLPS.

| Data | S1 | S2 | COD | MP1 | MP2 | o/slips | Warnings |
|------|------|------|------|------|------|---------|----------|
| 1(A) | **36.9** | 37.2 | **86** | 0.02 | 0.03 | **114** | S1, COD, o/slips |
| 2(B) | **38.5** | 35.2 | 97 | 0.05 | 0.08 | **253** | S1, COD, o/slips |
| 3(C) | **33.8** | **30.6** | **89** | 0.05 | 0.04 | **20** | S1, S2, COD, o/slips |
| 4(A) | **37.7** | 36.6 | **94** | 0.06 | 0.06 | **39** | S1, COD, o/slips |
| 5(B) | **42.9** | 37.6 | 100 | 0.02 | 0.03 | 1818 | S1 |
| 6(C) | **32.1** | **28.2** | **85** | 0.1 | 0.02 | **59** | S1, S2, COD, o/slips |
| 7(A) | **38.9** | 37 | **86** | 0.02 | 0.02 | **602** | S1, COD, o/slips |
| 8(B) | **35.5** | **34.4** | **85** | 0.03 | 0.04 | **257** | S1, S2, COD, o/slips |
| 9(C) | **36.7** | **33.6** | 98 | 0.27 | 0.43 | **40** | S1, COD, o/slips |
| 10(A) | **38.5** | 38.9 | 95 | 0.04 | 0.05 | **370** | S1, o/slips |
| 11(B) | **37.1** | 47.3 | **84** | 0.06 | 0.05 | **127** | S1, COD, o/slips |
| 12(C) | **34.3** | 41.2 | **87** | 0.05 | 0.03 | **342** | S1, COD, o/slips |
| 13(A) | **39.2** | 39.0 | 96 | 0.03 | 0.04 | **220** | S1, o/slips |
| 14(B) | **38.1** | 48.1 | **89** | 0.03 | 0.06 | **234** | S1, COD, o/slips |
| 15(C) | **32.1** | 42.2 | **93** | 0.04 | 0.05 | **352** | S1, COD, o/slips |

The warning indicators are in bold in the table.

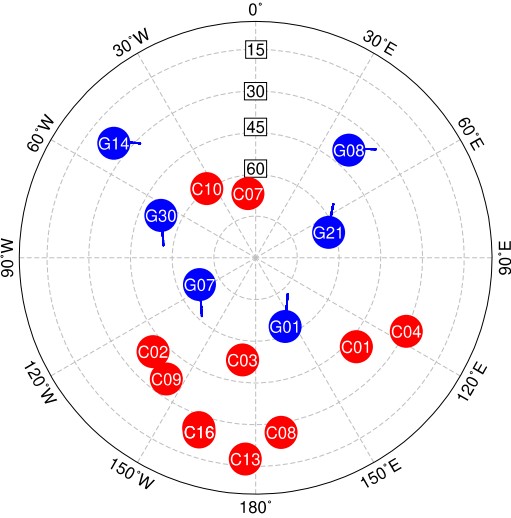

**Figure 10.** Skyplot of site A.

## 5. Discussion

To assess the ambiguity resolution and positioning performance with and without the static level constraints, the GNSS data for the first group in Section 4.1 was processed with and without combining the static level data. In the GNSS data processing, site A was set to be the reference station, and B and C were the monitoring stations. Because the baselines were short with about 1.5 m, we searched and solved the ambiguities for every epoch, and only single frequency data was processed. The detailed data processing strategies are shown in Table 1. The flow chart of data processing and analysis is shown in Figure 11. After getting GNSS and Static level observations, the time synchronization need to be considered, and then we processed the GNSS data with and without considering the constraint of static level observation.

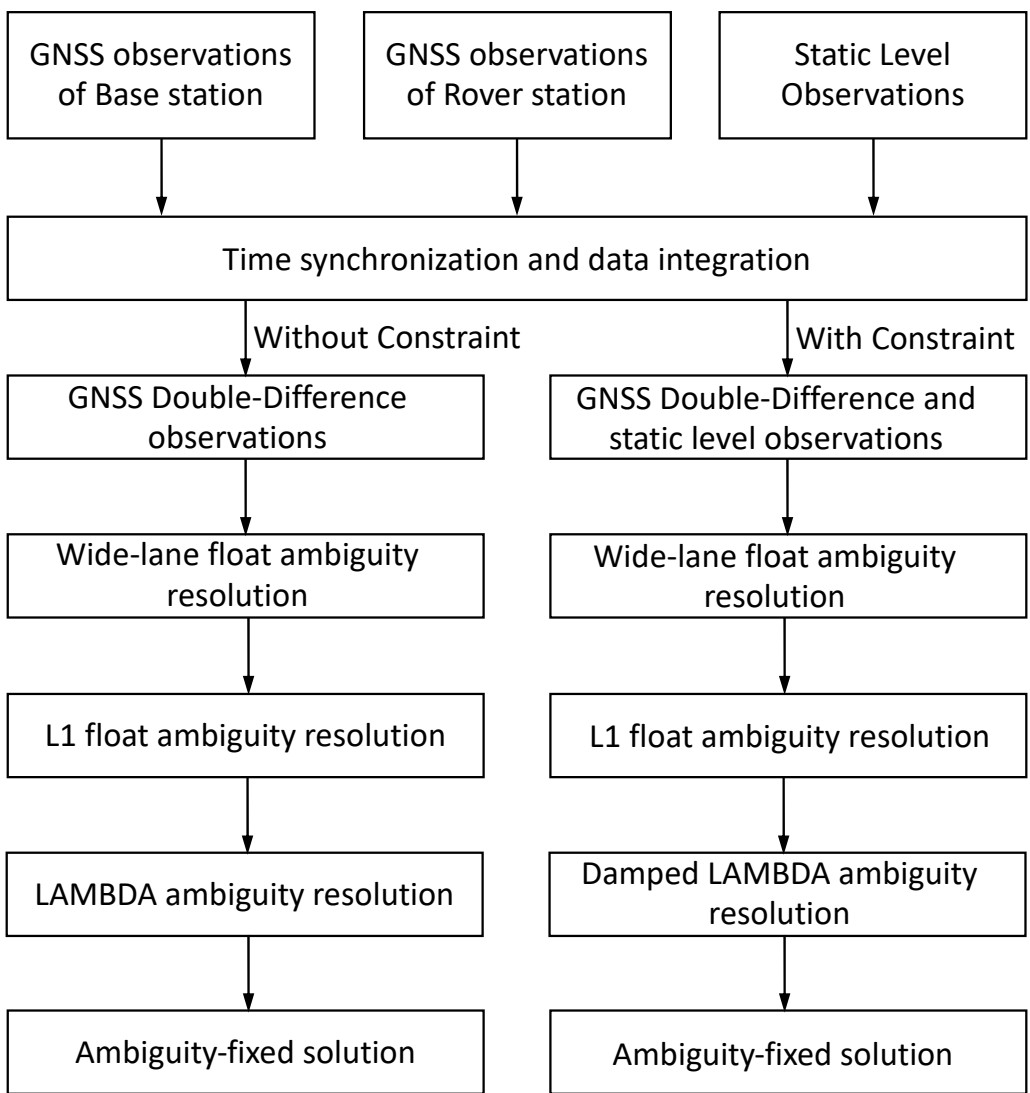

**Figure 11.** Flow chart of data processing.

Figure 12 shows the ambiguity resolution ratio test for the baseline A–B and A–C with and without static level constraints for an ambiguity resolution case. It can be seen that the time series with and without constraints have a large similarity for both baselines. However, the ratio averages of the baseline A–B and A–C are improved. Table 3 shows the ratio statistics with and without static level observation constraints. It shows that, with static level constraints, the ratio averages improvement can achieve from 20% to 30%. Although the ratios were large enough to fix the ambiguity for GNSS observation

only, the ratio improvements would help increase the ambiguity fixing rate in harsh observing environment.

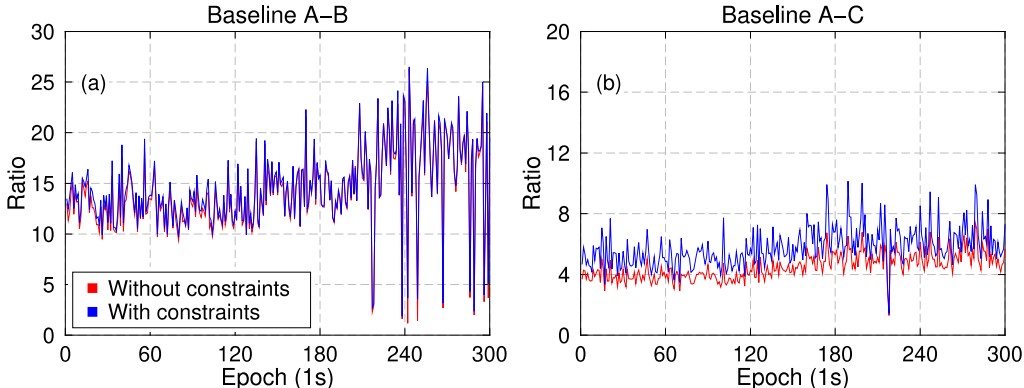

**Figure 12.** Ratio time series of baseline A–B (**a**) and A–C (**b**) with and without the static level constraints.

**Table 3.** Ratio statistics with and without static level observation constraints.

| Baseline | Session | Ratio without Constraints | Ratio with Constraints | Improvement (%) |
|---|---|---|---|---|
| A–B | 1 | 11.8 | 14.7 | 24.7% |
| A–B | 2 | 10.6 | 13.6 | 28.3% |
| A–B | 3 | 9.5 | 12.3 | 29.5% |
| A–B | 4 | 11.6 | 15.1 | 30.2% |
| A–B | 5 | 13.2 | 15.9 | 20.5% |
| A–B | 6 | 10.2 | 13.2 | 29.4% |
| A–C | 1 | 4.3 | 5.6 | 28.7% |
| A–C | 2 | 6.5 | 8.2 | 26.2% |
| A–C | 3 | 3.2 | 4.1 | 28.1% |
| A–C | 4 | 5.5 | 7.0 | 27.3% |
| A–C | 5 | 4.4 | 5.7 | 29.5% |
| A–C | 6 | 3.8 | 4.7 | 23.7% |

Figures 13–16 give the widelane and L1/B1 ambiguity float solution time series of baseline A–B and A–C with and without static levelling constraint, respectively. It can be seen that, the accuracy of widelane ambiguity float solution can be significantly improved with the static level constraint, from about 1 m to less than 0.5 mm. With the constraints of static level, the precision of the GNSS L1/B1 float solution is significantly improved in 3D directions, from 0.051 m to 0.015 m for A–B and 0.156 m to 0.041 m for A–C. Due to the constraint of static level, the precision of float ambiguity and solution is greatly improved, which would greatly speed up the fixing of ambiguity.

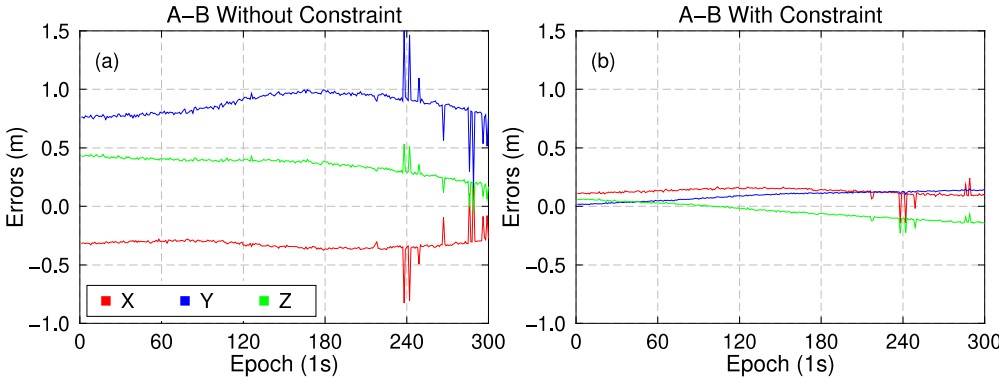

**Figure 13.** Widelane ambiguity float solutions of baseline A–B without (**a**) and with (**b**) static levelling constraint.

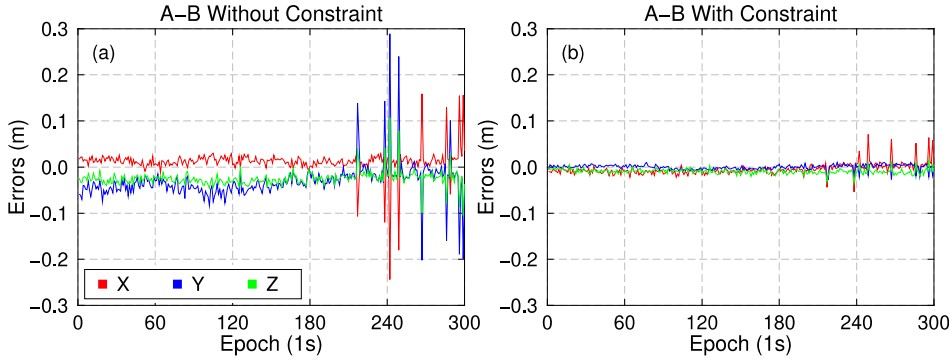

**Figure 14.** L1/B1 ambiguity float solutions of baseline A–B without (**a**) and with (**b**) static levelling constraint.

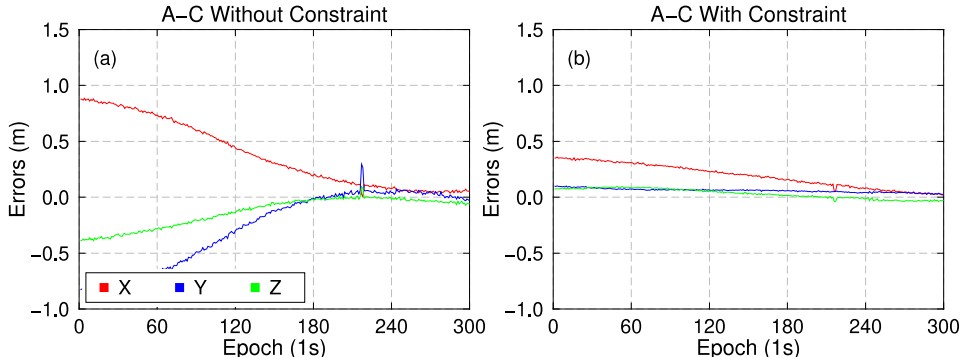

**Figure 15.** Widelane ambiguity float solutions of baseline A–B without (**a**) and with (**b**) static levelling constraint.

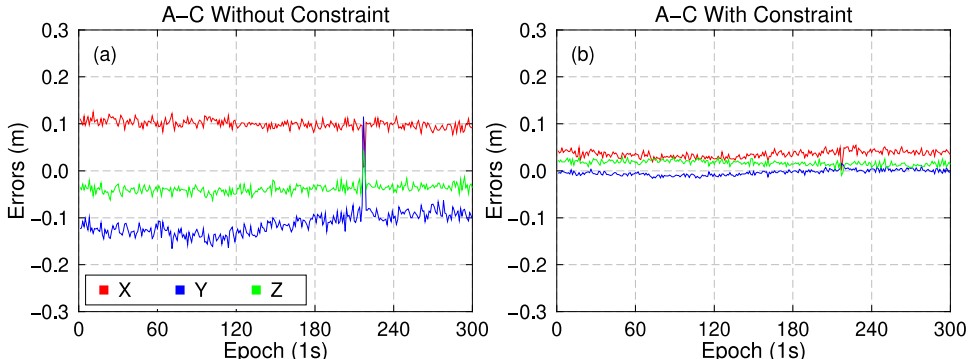

**Figure 16.** L1/B1 ambiguity float solutions of baseline A–C without (**a**) and with (**b**) static levelling constraint.

To further evaluate the accuracy of the fixed solutions, the fixed solutions with and without constraint were firstly transformed into the local cartesian coordinate system, where A was the origin. Then, the baseline solutions of A–B and A–C in North, East and Up directions were compared with the known baseline information to give the Root Mean Square (RMS) statistics in Table 4. It shows that, with only GNSS measurements observed for about 5 min, the positioning accuracy could achieve to 1 cm in horizontal direction and about 2 cm in vertical direction, due to large observation noise and signal sheltering by surroundings, even with the ambiguity-fixed solutions. However, with the static level constraint, the positioning accuracy in U direction was significantly improved from about 2 cm to better than 2 mm, which is even better than the accuracy in horizontal directions with about 3–6 mm with the static level constraint. The positioning accuracy is significantly improved with the constraint of static level observation.

**Table 4.** RMS statistics of the baseline resolution with and without constraint.

| Baseline | Sessions | RMS (m) | | | | | |
| | | E | | N | | U | |
| | | Without | With | Without | With | Without | With |
|---|---|---|---|---|---|---|---|
| A–B | 1 | 0.008 | 0.003 | 0.009 | 0.006 | 0.020 | 0.002 |
| A–B | 2 | 0.006 | 0.002 | 0.008 | 0.006 | 0.018 | 0.001 |
| A–B | 3 | 0.007 | 0.003 | 0.005 | 0.003 | 0.014 | 0.001 |
| A–B | 4 | 0.008 | 0.004 | 0.007 | 0.004 | 0.022 | 0.000 |
| A–B | 5 | 0.004 | 0.002 | 0.006 | 0.003 | 0.015 | 0.001 |
| A–B | 6 | 0.009 | 0.003 | 0.007 | 0.005 | 0.011 | 0.001 |
| A–C | 1 | 0.008 | 0.003 | 0.006 | 0.005 | 0.019 | 0.000 |
| A–C | 2 | 0.009 | 0.004 | 0.010 | 0.006 | 0.015 | 0.001 |
| A–C | 3 | 0.008 | 0.002 | 0.008 | 0.005 | 0.021 | 0.002 |
| A–C | 4 | 0.006 | 0.002 | 0.008 | 0.006 | 0.013 | 0.001 |
| A–C | 5 | 0.007 | 0.003 | 0.006 | 0.004 | 0.016 | 0.000 |
| A–C | 6 | 0.008 | 0.004 | 0.009 | 0.006 | 0.017 | 0.001 |

## 6. Conclusions

In this study, we proposed a method of integrating GNSS and static level observations tightly to enhance the ambiguity resolution and the positioning accuracy performance. The conclusions are shown in the following:

1. The vertical direction measurement results of static level can directly improve the vertical monitoring accuracy of the house monitoring to less than 1 mm.

2. The monitoring and observation environment of dilapidated houses is generally densely built urban areas, which may be affected by signal interference, occlusion, observation noise and other adverse factors in varying degrees. In the simulation experiment of this paper, quality inspection software is used to analyze the observation data quality of GNSS. The results show that in each observation period, there is a certain degree of early warning in many aspects such as SNR, cycle skip and data availability, indicating that the quality of GNSS data is not very ideal and may bring disadvantages in the application of high-precision monitoring.

3. The ambiguity resolution performance can be improved by incorporating the measurement of static level into GNSS positioning equation as external constraints. The accuracy of widelane ambiguity float solution can be significantly improved with the static level constraint, from about 1 m to less than 0.5 mm. With the constraints of static level, the precision of the GNSS L1/B1 float solution is significantly improved in 3D di-rections, from 0.051 m to 0.015 m for A–B and 0.156 m to 0.041 m for A–C. Due to the con-straint of static level, the precision of float ambiguity and solution is greatly improved, which would greatly speed up the fixing of ambiguity.

4. With only GNSS measurements observed for about 5 min, the positioning accuracy could achieve to 1 cm in horizontal direction and about 2 cm in vertical direction, due to large observation noise and signal sheltering by surroundings, even with the ambiguity-fixed solutions. The static level constraint can further improve the accuracy of the fixed solution from about 2 cm to better than 2 mm in vertical direction, which is even better than the accuracy in horizontal directions with about 3–6 mm with the static level constraint.

To sum up, although GNSS technology is an excellent choice in the application of automatic dangerous house monitoring, there will undoubtedly be some bottlenecks in the application of single GNSS system considering the application environment of dangerous house monitoring and the poor positioning accuracy of GNSS in the elevation direction. The multi-sensor fusion positioning has a broad prospect in the displacement monitoring of dilapidated house and other buildings, bridges etc., and the static level on the height direction of high precision measurements can be perfect to make up for the problem of insufficient accuracy of GNSS elevation. However, basically just simple fusion on the two independent monitoring results, called loose combination, is generally applied in the engineering applications. This paper explores the static level and GNSS "tight combination"

hardware integration and data processing method. Through the simulation experiment, it proves that the static level besides and can provide higher elevation accuracy, and can also serve as the constraint conditions of auxiliary GNSS high-accuracy. On the one hand, the static level improves the float solution and fixed solution precision of GNSS. On the other hand, it can improve the efficiency and reliability of GNSS ambiguity fixing.

It should be noted that, the experiment hardware design integration baseline is very short. In the absence of the static level constraints, the ambiguity of GNSS can be fixed with single epoch observations. Therefore, the performance of ambiguity fixing efficiency with static level constraints has not been fully reflected. More research especially for long baseline tests would be done in the future works. In addition, the elevation direction constraint provided by static level and the improvement of GNSS plane accuracy will be theoretically deduced and demonstrated in the next research.

**Author Contributions:** Conceptualization, J.Y. and W.X.; methodology, J.Y.; software, W.T.; validation, W.T.; formal analysis, R.X.; investigation, R.X.; resources, W.X.; data curation, W.T.; writing—original draft preparation, J.Y.; writing—review and editing, W.X.; visualization, W.X.; supervision, W.T.; project administration, J.Y.; funding acquisition, J.Y. All authors have read and agreed to the published version of the manuscript.

**Funding:** This research was funded by the Fundamental Research Funds for the Central Universities, grant number WUT: 2022IVA035.

**Data Availability Statement:** The experimental data could provide by the corresponding author upon request.

**Conflicts of Interest:** The authors declare that they have no known competing financial interest or personal relationships that could have appeared to influence the work reported in this paper.

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
