# Peer review of "Tight Integration of GNSS and Static Level for High Accuracy Dilapidated House Deformation Monitoring"

_remotesensing, doi:10.3390/rs14122943_

Round 1

Reviewer 1 Report

The paper is well written and easy to read. An algorithm of tightly integrating the GNSS with the static level is proposed, and the results presented are good, which has proven its application value in deformation monitoring of the dilapidated house. However, some minor revisions also need to be addressed before publication.

1. Lines 154-156, this paragraph is incoherent with the context, and the author may have written it by mistake. Please delete it.

2. Line 158, the symbol “lambda” in Eq. (2) is not explained. As far as I know, it means the wavelength of the carrier phase measurement. But the problem is that the symbol “lambda” also denotes the longitude in Eqs. (3) and (4). Please use different symbols to distinguish them.

3. Line 158, the symbol “N” in Eq. (2) represents the ambiguity, while it represents the coefficient matrix of the normal equation in Eqs. (6) and (7). Please use different symbols to distinguish them.

4. Line 171, replace “X” with “x”.

5. Line 172, the resolution of the figure is too low. Furthermore, the origin of the geocentric cartesian coordinates should use other symbol rather than “O” to prevent confusion with station O.

6. Line 174, replace the sentence “If the latitude and longitude of …” with “If the longitude and latitude of …”

7. Line 176, replace “×” with “·” or just delete it.

8. Lines 198 and 201, the abbreviations “OTF” and “LAMBDA” need to be defined as they first appear in the text.

9. Lines 236-239, the “sampling rate” should be described as 1 Hz instead of 1 s, please check the whole manuscript.

Author Response

The paper is well written and easy to read. An algorithm of tightly integrating the GNSS with the static level is proposed, and the results presented are good, which has proven its application value in deformation monitoring of the dilapidated house. However, some minor revisions also need to be addressed before publication.

Reply: Many thanks for your valuable comments.

  1. Lines 154-156, this paragraph is incoherent with the context, and the author may have written it by mistake. Please delete it.

Reply: This paragraph has been removed. Thanks.

  1. Line 158, the symbol “lambda” in Eq. (2) is not explained. As far as I know, it means the wavelength of the carrier phase measurement. But the problem is that the symbol “lambda” also denotes the longitude in Eqs. (3) and (4). Please use different symbols to distinguish them.

Reply: The wavelength “lambda” has been modified to ‘’ in the equations and statements.

  1. Line 158, the symbol “N” in Eq. (2) represents the ambiguity, while it represents the coefficient matrix of the normal equation in Eqs. (6) and (7). Please use different symbols to distinguish them.

Reply: The symbol for normal equation has been modified to ‘’.

  1. Line 171, replace “X” with “x”.

Reply: The statement has been modified.

  1. Line 172, the resolution of the figure is too low. Furthermore, the origin of the geocentric cartesian coordinates should use other symbol rather than “O” to prevent confusion with station O.

Reply: The figure has been re-plotted and the station symbol has been modified to P in the Figure 4 and the statements.

  1. Line 174, replace the sentence “If the latitude and longitude of …” with “If the longitude and latitude of …”

Reply: The statement has been modified. Many thanks.

  1. Line 176, replace “×” with “·” or just delete it.

Reply: The equation has been modified. Thanks.

  1. Lines 198 and 201, the abbreviations “OTF” and “LAMBDA” need to be defined as they first appear in the text.

Reply: The full names of these abbreviations have been given in the revised manuscript. Thanks.

  1. Lines 236-239, the “sampling rate” should be described as 1 Hz instead of 1 s, please check the whole manuscript.

Reply: The statement has been modified.

Reviewer 2 Report

Dear Authors,

very interesting work you've done.

Several minor things I've noticed in the text to be corrected:

line 104 - lower case of "b" in Bridges,

line 153 - I think after this line equations (1) and (2) should be written,

line 198 - declare the On The Fly (OTF) acronym and

line 201 - declare LAMBDA acronym.

In sub-section 4.3, please describe which software for GNSS processing did you use and which processing strategy did you apply.

Thank you.

With kind regards,

Reviewer

Author Response

Dear Authors,

very interesting work you've done.

Several minor things I've noticed in the text to be corrected:

Reply: Many thanks for reviewing our manuscript and giving valuable comments.

line 104 - lower case of "b" in Bridges,

Reply: The word has been modified. Thanks.

line 153 - I think after this line equations (1) and (2) should be written,

Reply: Lines 154-156 have been removed from the manuscript. Thanks.

line 198 - declare the On The Fly (OTF) acronym and line 201 - declare LAMBDA acronym.

Reply: The full names of these abbreviations have been given. Thanks.

In sub-section 4.3, please describe which software for GNSS processing did you use and which processing strategy did you apply.

Reply: The GNSS data processing software is a homemade software which is called GNSStrack. The detailed processing strategy is shown in Table 1.

Thank you.

With kind regards,

Reviewer